# Automatic Classification for Sagittal Craniofacial Patterns Based on Different Convolutional Neural Networks

**DOI:** 10.3390/diagnostics12061359

**Published:** 2022-05-31

**Authors:** Haizhen Li, Ying Xu, Yi Lei, Qing Wang, Xuemei Gao

**Affiliations:** 1Department of Orthodontics, Stomatology School and Hospital of Peking University, Beijing 100081, China; 1911110543@pku.edu.cn (H.L.); 2111110550@stu.pku.edu.cn (Y.X.); 2School of Software Engineering, Faculty of Information Technology, Beijing University of Technology, Beijing 100124, China; leiyi9345@163.com; 3Department of Automation, Tsinghua University, Beijing 100084, China; 4Pharmacovigilance Research Center for Information Technology and Data Science, Cross-Strait Tsinghua Research Institute, NO.516 Qishan North Road, Huli District, Xiamen 361000, China

**Keywords:** artificial intelligence, orthodontic sagittal skeletal pattern classification, convolutional neural networks

## Abstract

(1) Background: The present study aims to evaluate and compare the model performances of different convolutional neural networks (CNNs) used for classifying sagittal skeletal patterns. (2) Methods: A total of 2432 lateral cephalometric radiographs were collected. They were labeled as Class I, Class II, and Class III patterns, according to their ANB angles and Wits values. The radiographs were randomly divided into the training, validation, and test sets in the ratio of 70%:15%:15%. Four different CNNs, namely VGG16, GoogLeNet, ResNet152, and DenseNet161, were trained, and their model performances were compared. (3) Results: The accuracy of the four CNNs was ranked as follows: DenseNet161 > ResNet152 > VGG16 > GoogLeNet. DenseNet161 had the highest accuracy, while GoogLeNet possessed the smallest model size and fastest inference speed. The CNNs showed better capabilities for identifying Class III patterns, followed by Classes II and I. Most of the samples that were misclassified by the CNNs were boundary cases. The activation area confirmed the CNNs without overfitting and indicated that artificial intelligence could recognize the compensatory dental features in the anterior region of the jaws and lips. (4) Conclusions: CNNs can quickly and effectively assist orthodontists in the diagnosis of sagittal skeletal classification patterns.

## 1. Introduction

Recently, artificial intelligence (AI) technology based on convolutional neural networks (CNNs) has emerged as an efficient and reliable tool for medical imaging diagnostics [1,2,3]. CNNs can independently provide auxiliary diagnoses for doctors by automatically learning from a large number of labelled medical images [4]. It is a fact that the application of CNNs is changing medical diagnostics, as they can assist doctors by improving diagnostic accuracy and increasing effectiveness [5]. Many scholars have tried to use CNNs for image auxiliary diagnosis in the field of orthodontics. Seo used CNNs to classify the cervical vertebral maturation stages on 600 lateral cephalometric radiographs [6]. Yoon used cascaded CNNs for landmark detection in cephalometric analyses with a database of 600 samples [7]. CNNs have been proven to have remarkable potential for assisting in many orthodontic diagnoses processes, and more research should be carried out [8].

As a fundamental part of the orthodontic diagnosis process, sagittal skeletal pattern classification is of great importance to orthodontists for formulating treatment plans and predicting facial growth [9]. Sagittal skeletal pattern diagnosis is a complex process because it usually involves a combination of indices, rather than merely distinguishing the boundary line. Traditionally, the diagnosis of sagittal skeletal patterns relies on several cephalometric measurements, such as the ANB angle [10], Wits value [11], APDI index [12], and Beta angle [13]. However, cephalometric analysis is a time-consuming and laborious process that calls for professional training and repeated practice. Besides, errors and bias are inevitable, due to the subjective nature of defining the landmarks manually. CNNs are an alternative method for sagittal skeletal classification because they are especially suitable for overcoming the limitations associated with manual identification methods.

In fact, an attempt to apply CNNs to sagittal skeletal pattern classification was made by Yu et al., in 2019. They successfully trained a CNN model with more than 90% accuracy, which proved the effectiveness of CNNs for sagittal skeletal classification [14]. Moreover, with rapid developments in the field of computer vision, algorithm engineers have developed multiple CNN model architectures, in order to accomplish different tasks. However, how to choose an appropriate CNN model is a puzzling problem for doctors when applying CNNs to the actual clinical work. Thus, it is imperative to learn the differences between model performances and the characteristics of CNNs. No comparative studies have been conducted on using different CNNs to classify sagittal skeletal patterns, although they could help guide the application of CNNs and support the strive for methodological standardization.

After the literature review, we realized that orthodontists do not need to understand the internal structure of CNNs; it is too complex for them to understand. What orthodontists need to do is to choose a suitable model from the many CNNs and validate its accuracy. In order to better apply the CNN model to clinical work, we believe it is vital to compare the performance of different CNN models. This present study aims to train and compare the model performances of different CNN algorithms for classifying sagittal skeletal patterns on lateral cephalometric radiographs.

## 2. Materials and Methods

This retrospective cross-sectional study was approved by the Institutional Review Board of Peking University School of Stomatology (PKUSSIRB-202054025). The design and implementation of the study strictly followed the principles of the Declaration of Helsinki, as well as its appendix. The schematic presentation of the overall methodology is briefly illustrated in Figure 1a.

### 2.1. Data Collection

A total of 2432 cephalometric lateral radiographs were collected from the Department of Orthodontics at Peking University Hospital of Stomatology between 2017 and 2020. The radiographs that fulfilled the following criteria were included in the study: (1) permanent dentition; (2) lateral cephalometric radiographs taken before orthodontic treatment, with Frankfort horizontal plane parallel to the ground and in the maximal intercuspal position. The exclusion criteria were as follows: (1) fuzzy images or images with other quality problems; (2) poor superimposition of bilateral anatomic structures; and (3) with previous maxillofacial trauma or surgery. The patients included 1018 males and 1413 females, aged between 12 to 42 years. The mean age was 25.4 ± 4.3 years. All the cephalometric lateral radiographs were captured using a scanner, i.e., the Veraviewepocs 2D (J Morita Corp, Kyoto, Japan), with the following parameters: scanning time, 4.9 s; tube current, 5–10 mA; tube voltage, 90 kV. All the images were stored in JPG format.

### 2.2. Data Labeling and Dataset

In this research, the samples were divided into three categories based on the Chinese normal mean value of ANB and Wits [15,16]: skeletal Class I pattern (5° ≥ ANB ≥ 0° and 2 ≥ Wits ≥ −3), skeletal Class II pattern (ANB > 5° and Wits > 2), and skeletal Class III pattern (ANB < 0° and Wits < −3). Each image was classified into one of the three classes after the manual measurements. Samples that failed to meet the criteria were excluded.

In this study, the cephalometric analysis was completed by the same senior orthodontist, who was asked to perform the analysis twice, with an interval of one week. A total of 100 lateral cephalometric films were selected for a pre-experiment, in which the ANB angles and Wits values were measured repeatedly. The intra-class correlation coefficient (ICC) of the two measurements was calculated, and the ICCs for the ANB angle and Wits value were 0.965 and 0.934, respectively.

The images of the three classes were further divided into three groups: the training set for model training, validation set for tuning hyper-parameters of the CNNs, and testing set for the final evaluation of its classification ability, according to the percentages of 70%, 15%, and 15%. The details of the dataset are given in Table 1.

### 2.3. Data Processing

There were two steps in the image preprocessing, which are shown in Figure 1b. The first step was to crop the relative region in the cephalometric radiographs and remove the excessive parts that were not conducive to the classification of sagittal skeletal patterns, such as medical record information, positioning frames, and radiation protection devices. This step was carried out with the Photoshop software. In the second step, the images were resized into 224 × 224 pixels using the OpenCV package. 

### 2.4. Data Augmentation

To avoid overlearning on small data sets, the following data augmentation techniques were also employed in this research [17]: random rotation, random scaling, random translations, and random changes in contrast and brightness. The schematic diagram of the data augmentation process is shown in Figure 1c. Half of the samples were randomly selected to perform data augmentation in each training epoch, resulting in 121,600 (1216 × 100) new data after 100 epochs of training.

### 2.5. Convolutional Neural Network Constriction

Medical data usually utilize the strategy of transfer learning [18], that is, using models that have been trained on non-medical data. Four representative CNNs, which were validated via ImageNet Large-Scale Visual Recognition Competition [19], were selected as candidates for our study: VGG16 [20], GoogLeNet [21], DenseNet161 [22], and ResNet152 [23]. We retrieved the pre-trained models of the four candidate CNN models from the PyTorch zoo (https://pytorch.org/ accessed on 17 January 2022). The only change we made to the CNNs was to change the three neurons in the output layer, so that they could correspond to the three sagittal skeletal patterns.

### 2.6. Training Details and Strategy

All the layers of the CNN model were trained and upgraded using fine-tuning techniques [24]. Each of CNN models in this study were trained for 100 epochs using a stochastic gradient descent (SGD) optimizer after initializing the parameters of the pre-trained models. The hyper-parameters of the CNNs were adjusted according to their model performance on the validation set for many times of training. At last, an individualized hyper-parameter combination, including the learning rate, batch size, momentum, and weight decay, was determined, in order to maximize the capabilities of the CNNs. All the training processes of this study were performed on the server of the computing platform of Peking University Hospital of Stomatology, with the NVIDIA Tesla P100 graphic processing unit.

### 2.7. Model Testing and Evaluation Metrics

For each of the four CNNs, the model with the lowest loss in the validation set was selected to verify the final model performance for the test set. The confusion matrix was used to describe the results of the CNN model that was developed for sagittal skeletal pattern classification. To evaluate the classification performance of the model for the three subgroups, the following indicators were calculated, according to the confusion matrix: precision, recall rates, F1 score, and classification accuracy. The receiver operating characteristic (ROC) curves were plotted, in order to describe the classification ability of CNNs, and the AUC values were also calculated. 

### 2.8. Model Visualization

In this study, the visualization of the CNNs was achieved using class activation mapping (CAM) technology [25]. We generated a heat map that could highlight the discriminative regions, in order to identify the special categories (Figure 1d).

## 3. Results

The model performances of the four CNNs during the training stage are shown as the accuracy-epoch and loss-epoch curves in Figure 2. All four CNNs achieved good convergence results after 100 epochs of training without overfitting.

The model size, training time, inference time, classification accuracy, and AUC value of the CNNs are shown in Table 2. The four CNNs achieved good classification performances in the test set, with all accuracies above 87% and an average AUC above 0.95. The accuracies of the four CNNs were ranked as follows: DenseNet161 > ResNet152 > VGG16 > GoogLeNet. The ranking of the calculation speed was GoogLeNet > VGG16 > DenseNet161 = ResNet152. The ranking of the model size was VGG16 > ResNet152 > DenseNet161 > GoogLeNet.

The detailed results of the test set are shown as multi-class confusion matrices in Figure 3. The majority of the samples were located diagonally from the confusion matrix. The accuracy, recall, and F1 value of each subset were calculated based on the confusion matrix (Table 3). In addition, ROC curves were drawn for each sagittal skeletal pattern corresponding to each CNN model, and then the AUC values were obtained (Figure 4). The F1 and AUC values for each subgroup suggest that all CNNs showed the highest accuracy in identifying the skeletal Class III pattern, followed by the skeletal Class II and I patterns.

The class activation map of the four CNNs is shown in Figure 5. Based on the direct visual analysis, we suggest that the main activation region of the four CNNs was located in the region of the anterior jaws and lips, indicating that there is no overfitting in the trained CNNs. However, the activation areas of the three sagittal skeletal patterns are a little different. The activation area of the skeletal Class I pattern is in the anterior alveolar region of the maxilla and mandible (Figure 5a–d). However, the activation area of the skeletal Class II pattern was closer to the mandible, and the skeletal Class III pattern was closer to the maxilla.

The wrongly classified samples are shown in Table 4. The ANB angels of these misjudged skeletal Class I and II samples were near 5°, and those of the misjudged skeletal Class I and III samples were near 0°. It was inferred that the CNN models were limited in classifying patients in the boundary state among the three categories.

## 4. Discussion

Automatic diagnosis based on AI technology is gaining extensive attention as a practical clinical auxiliary diagnosis tool, and it is a developing trend in orthodontics [26]. In order to achieve a better application of CNNs, we trained four representative CNNs and compared their model performance in this research. We found that all four CNNs could easily be trained in less than 40 min under the condition of transfer learning. It can be seen that the four candidate CNN models are capable of classifying the sagittal skeletal patterns to a certain degree, with all the accuracies exceeding 87%. However, the classification ability, model size, and inference speed of the four CNN algorithms are slightly different, with DenseNet161 having the highest accuracy and GoogLeNet possessing the smallest model size and fastest inference speed. In summary, our research indicates that AI technology based on CNNs is a fast, accurate, and general method for sagittal skeletal pattern classification, thereby promoting the application of deep learning technology in the field of orthodontics.

In this study, CNNs exhibited three advantages for orthodontic imaging analysis. First, CNNs judged the sagittal skeletal patterns according to the overall features of the lateral cephalometric radiographs. This method can effectively avoid the shortcomings that are inherent in cephalometric analysis based on either angular or linear measurements, as has been described by Moyers [27]. Second, deep CNNs are regarded as a universal approximation machine learning algorithm [28]. Given a set of enough imaginary data, CNNs can automatically learn the laws of the data and make independent decisions by using the end-to-end training method. The generality of CNNs makes it easy for orthodontists to train their own CNN models for customized diagnostic standards. Third, even the slowest inference speed of the four CNN models in our study reached 0.32 s/per film, significantly improving the efficiency of orthodontic diagnoses. In summary, CNNs are expected to be applied in clinical practice on a large scale.

All four candidate CNN models showed good classification accuracy in this study, with a ranking of DenseNet161 > ResNet152 > VGG16 > GoogLeNet. The literature review suggests that the differences in the model performances among the CNNs are related to their internal structures. GoogLeNet can save hardware resources because its network structure only has 22 layers. Two efforts have been made to reduce the model parameters. One is to use sparse connections to overcome the problem of information redundancy, and the other is to use a global average pooling layer, instead of full-connection layers, which could effectively reduce the connection density. The application of smaller models will assist in making AI technology more sustainable [2]. The ResNet152 model improved performance by establishing a skip connection between the front and back layers, which was helpful to the back propagation of the gradient in the training process, so as to train a deeper CNN network. The VGG16 had the largest model size among the four models, owing to its full-connection layers. However, as evidenced by the fact that its classification accuracy only ranked third, this structure seemed to have little effect on improving the model performance. DenseNet161 achieved the highest accuracy of 89.58% and an AUC value of 0.977, since it increased the reuse of the features through the dense connection among all layers and with the information flow maximized. It is worth mentioning that reuse of the features did not significantly increase the model size of DenseNet161, which facilitated its clinical application. Based on the results of this study, it can be concluded that each of these four models has its own characteristics and can be selected tendentiously for different application scenarios. This study also found that the model performance of different CNNs is gradually increasing, with continuous advancements in the internal structures of CNNs, indicating that advances in computer algorithms have contributed to the improvement of model performances.

The black-box nature of CNNs has restricted their clinical use, as they do not provide any explanation or knowledge as to why or how they make predictions [26]. Recent studies on medical imaging analyses have placed a greater emphasis on the interpretability of AI models because a medical diagnosis system needs to be transparent, understandable, and explainable to gain the trust of physicians [29]. Our results demonstrate that the activation area of the four CNNs was mainly located in the lower part of the cephalometric images and did not detach from the anterior jaw region, thus indirectly supporting the absence of overfitting in our models. From the activation map, it can be concluded that machine learning not only focused on the sagittal relationship of the maxilla and mandible but also paid particular attention to the compensatory features of teeth. Additionally, it is suggested that the way CNNs classify the sagittal skeletal pattern is similar to that of orthodontists, which makes CNNs more acceptable to orthodontists.

As we know, CNNs mainly classify images according to geometric features [3], so the differences in the model performances among the three sagittal skeletal patterns may result from their different features. In general, patients with skeletal class III often have a compensatory inclination of maxillary incisors and lingual inclination of mandibular incisors [30], while patients with class II often have a compensatory labial inclination of mandibular incisors and small chin [31]. In this study, the classification abilities of the CNNs for the subgroups of sagittal skeletal patterns were ranked as follows: Class III > Class II > Class I, which may result from the fact that the image features of Classes III and II are more distinct than that of Class I. The typical features within the Class III and II images were more easily captured by the CNNs.

Many previous studies have put the spotlight on the classification accuracy of the analysis model. In this study, we further analyzed the misjudged samples of CNNs, in order to determine their shortcomings. We found that CNNs performed poorly when distinguishing between samples in the boundary state of the three sagittal skeletal patterns, and further training of these samples should be undertaken in the future.

As a potential limitation of this study, the CNN models we used were pre-trained and validated with the public dataset, rather than customized models for sagittal skeletal pattern classification. Individually constructed CNN architectures should be exploited in our continual research. Besides, we have not conducted large-scale clinical trials for the CNN models, and the final generalization ability of the model remains undetermined.

## 5. Conclusions

Four CNN models, namely DenseNet161, ResNet152, GoogLeNet, and VGG16, have a good ability to classify sagittal skeletal patterns. The model performances of the CNNs are different, with DenseNet161 having the highest accuracy and GoogLeNet possessing the smallest model size and fastest inference speed. The classification ability of all CNNs for sagittal skeletal patterns also differs, ranking: Class III > Class II > Class I. Most of the samples that were misclassified by the CNNs were boundary cases.

## Figures and Tables

**Figure 1 diagnostics-12-01359-f001:**
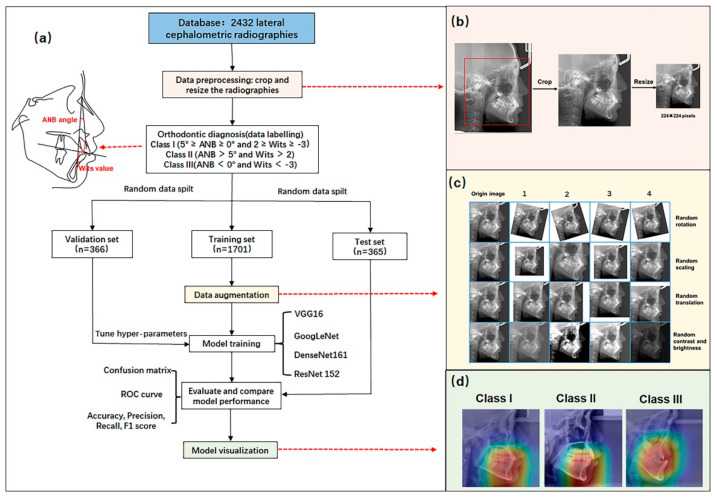
(**a**) Flow chart of the experimental process. (**b**) Schematic diagram of data preprocessing. (**c**) Schematic diagram of data augmentation. (**d**) Schematic diagram of model visualization.

**Figure 2 diagnostics-12-01359-f002:**
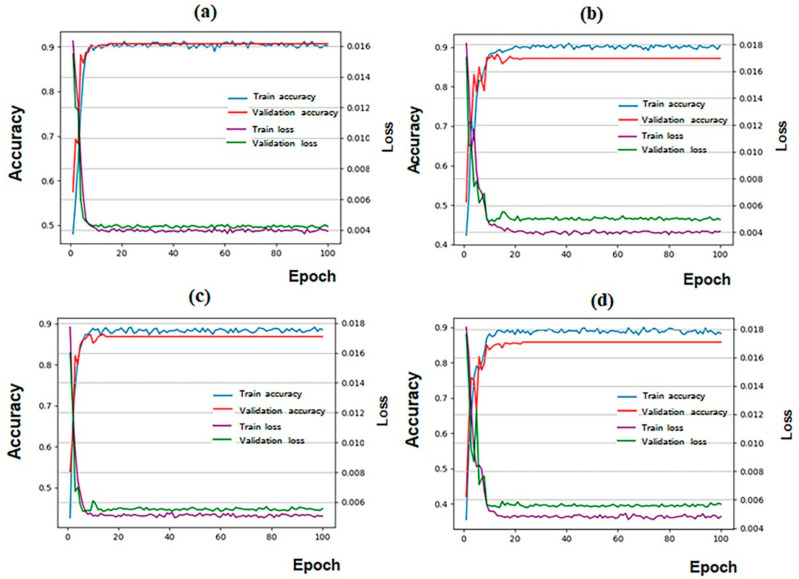
Training results for the four candidate CNN models. DenseNet161 had the highest accuracy. (**a**) DesneNet161; (**b**) ResNet152; (**c**) VGG16; (**d**) GoogLeNet. The horizontal axis of the graph represents the training epochs. The left vertical axis of the graph represents the classification accuracy, and the right vertical axis of the graph represents the loss value.

**Figure 3 diagnostics-12-01359-f003:**
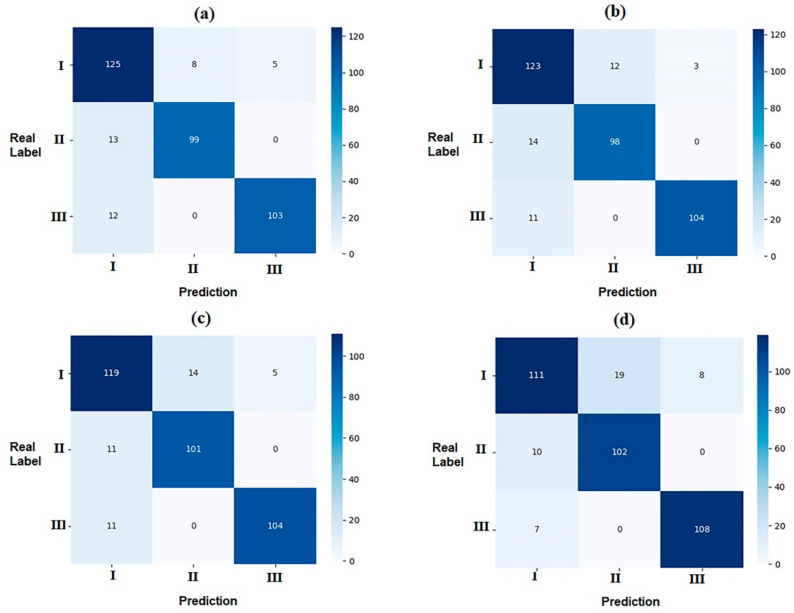
Confusion matrix of the four CNN models of the test set: data concentrating in the diagonal line indicates better predictive performance in the four models. (**a**) DesneNet161; (**b**) ResNet152; (**c**) VGG16; (**d**) GoogLeNet.

**Figure 4 diagnostics-12-01359-f004:**
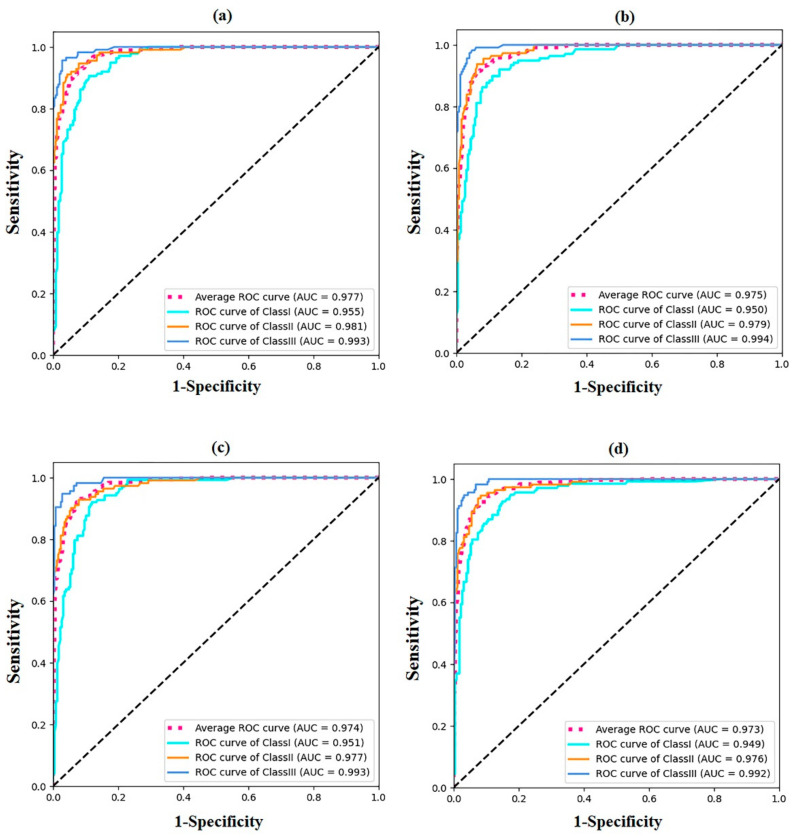
ROC curves of the four CNN models of the test sets. All four CNNs models have good ability to classify the sagittal skeletal patterns. (**a**) DesneNet161; (**b**) ResNet152; (**c**) VGG16; (**d**) GoogLeNet.

**Figure 5 diagnostics-12-01359-f005:**
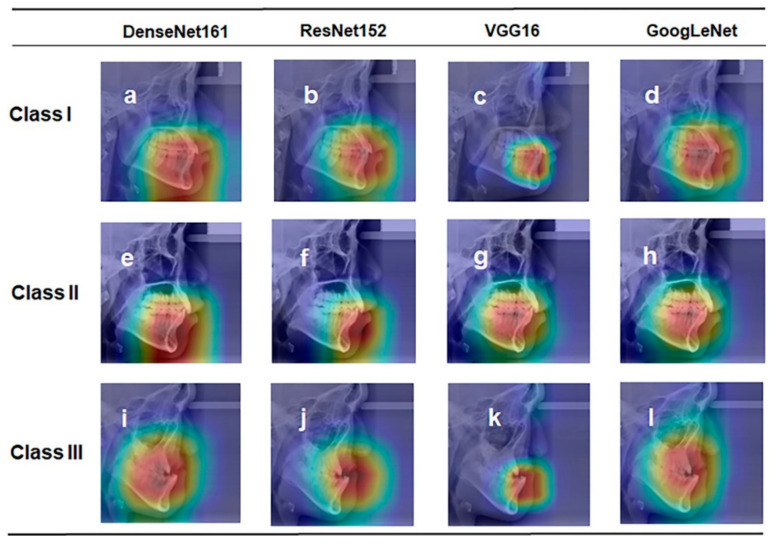
Representative class activation maps of different CNNs; (**a**–**d**) representative images of Class I; (**e**–**h**) representative images of Class II; (**i**–**l**) representative images of Class III.

**Table 1 diagnostics-12-01359-t001:** Numbers of patients assigned to the training, validation, and test sets of the three sagittal skeletal subgroups.

	Class I	Class II	Class III	Total
Training set	642	525	534	1701
Validation set	138	113	115	366
Test set	138	112	115	365
Total	918	750	764	2432

**Table 2 diagnostics-12-01359-t002:** Model size, training time, classification accuracy, inference time, and AUC value of the four CNNs.

	Model Size	Training Time (min)	Accuracy	Inference Time (s/per Image)	AUC Value
DenseNet161	102	40	89.58	0.32	0.977
ResNet152	222	38	89.04	0.32	0.974
VGG16	512	33	88.76	0.26	0.973
GoogLeNet	21.5	27	87.94	0.083	0.972

**Table 3 diagnostics-12-01359-t003:** Precision, recall rates, and F1 scores of four CNNs on the test set for each sagittal skeletal pattern subgroup.

	Precision (95% CI)			Recall (95% CI)			F1 Score (95% CI)		
	I	II	III	I	II	III	I	II	III
DenseNet161	0.83(0.77–0.88)	0.93(0.86–0.96)	0.95(0.90–0.98)	0.91(0.85–0.94)	0.88(0.81–0.93)	0.90(0.83–0.94)	0.87(0.81–0.691)	0.90(0.83–0.94)	0.92(0.86–0.96)
ResNet152	0.83(0.75–0.87)	0.89(0.82–0.94)	0.97(0.92–0.99)	0.89(0.83–0.93)	0.88(0.80–0.92)	0.90(0.84–0.95)	0.86(0.79–0.90)	0.88(0.81–0.93)	0.94(0.88–0.97)
VGG16	0.84(0.78–0.89)	0.88(0.81–0.93)	0.95(0.90–0.98)	0.86(0.79–0.91)	0.90(0.83–0.94)	0.90(0.84–0.95)	0.85(0.78–0.90)	0.89(0.82–0.93)	0.93(0.87–0.96)
GoogLeNet	0.87(0.80–0.92)	0.84(0.77–0.90)	0.93(0.87–0.96)	0.80(0.73–0.86)	0.91(0.84–0.95)	0.94(0.88–0.97)	0.83(0.76–0.89)	0.88(0.80–0.92)	0.94(0.87–0.96)

**Table 4 diagnostics-12-01359-t004:** The average ANB angles and Wits values of the misjudged samples by the four CNNs.

	DenseNet161		ResNet152		VGG16		GoogLeNet	
	ANB	Wits	ANB	Wits	ANB	Wits	ANB	Wits
I–II *	4.38 ± 0.34	1.63 ± 0.25	4.43 ± 0.31	1.45 ± 0.3	4.38 ± 0.29	1.51 ± 0.17	4.51 ± 0.29	1.48 ± 0.26
II–I *	5.61 ± 0.35	2.85 ± 0.48	5.45 ± 0.33	3.23 ± 0.57	5.57 ± 0.29	2.85 ± 0.61	5.63 ± 0.16	3.07 ± 0.61
I–III *	0.66 ± 0.34	−1.92 ± 0.64	0.50 ± 0.20	−1.83 ± 0.84	0.54 ± 0.30	−1.94 ± 0.32	0.44 ± 0.32	−1.79 ± 0.65
III–I *	−0.38 ± 0.31	−3.34 ± 0.32	−0.42 ± 0.29	−3.75 ± 0.29	−0.55 ± 0.30	−3.46 ± 0.22	−0.33 ± 0.22	−3.31 ± 0.23

* I–II represents the skeletal Class I samples misjudged as skeletal Class II, II–I represents the skeletal Class II samples misjudged as skeletal Class I, I–III represents the skeletal Class I samples misjudged as skeletal Class III, and III–I represents the skeletal Class III samples misjudged as skeletal Class I.

## Data Availability

All data generated or analyzed during this study are included in this published article.

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
