# Peer review of "Automatic Classification for Sagittal Craniofacial Patterns Based on Different Convolutional Neural Networks"

_diagnostics, 2022, doi:10.3390/diagnostics12061359_

Round 1

Reviewer 1 Report

This is a good manuscript with an interesting idea. However, the classification of facial types is not a challenging task that demands AI tools. Actually, it is the primary goal of any dental student to be able to recognize Class l, Class ll, and Class lll. For this reason, I would suggest the authors add repeatability of the classification by a human expert to compare with their CNN results (not only the ICC of the values), but the human judge should classify all the images and, after a month, repeat the process. That would add a great value to the orthodontics community in addition to the information from the CNN comparisons.

  1. “…CNNs can independently make a medical diagnosis in place of doctors by automatically learning from a large number of labelled medical images[4]…”

Can a CNN make a diagnosis without the need of a doctor? I do not believe so. It can help diagnose but not make a final decision, replacing the doctor.

  1. “….It is a fact that the application of CNNs is changing the way of medical diagnostics as they can improve diagnostic accuracy, eliminate differences between different doctors and increase the effectiveness…”

How can a CNN eliminate differences between doctors? Please, re-phrase sim.

  1. “…Although CNNs have been proven to hold the remarkable potential to assist orthodontics in many diagnosis processes, more relevant research has not been carried out[8]…”

This is too vague. What is relevant research means? In my opinion many relevant research has been published in this field.

  1. “…After the literature review, we hypothesized that the difference in model performance among CNNs is related to their internal structures. Despite the fact that orthodontists may not need to be conversant with the internal structure of CNNs, we believe it is vital to compare the performance of different CNN models when applied in clinical practice…”

I’m sorry, but I did not understand this statement, especially the relationship between orthodontists and CNN.

  1. Figure 1 needs more information in the legend. Please, add to the legend each step and letter (a,b,c,d and what that represents).
  2. “…orthodontic treatment with Frankfort horizontal plane parallel to the ground and in the maximal intercostal position…”

Typo here.

  1. “…The 87 patients included 1018 males and 1413 females, aged from 12 to 42 years. Their average age was 25.4±4.3 years old…”

Another English problem here.

  1. “…In this research, the samples were divided into three categories based on the Chinese normal mean value of ANB and Wits…”

Reference?

  1. Why did you choose 70, 15, and 15, and not any other value? Please, explain your text also.
  2. Data processing: Please, provide the estimated number of hours spent during this manual processing in photoshop and Open-CV package. Also, explain why you had to do this additional step.
  3. Table 3: Please, provide the SD in addition to the values.
  4. “…The class activation map of the four CNNs is shown in Fig 5. The main activation region of four CNNs was located in the region of the anterior jaws and lips, indicating that there is no overfitting in the trained CNNs. However, the activation areas of the three sagittal skeletal patterns are a little different. The activation area of skeletal Class I pattern is in the anterior alveolar region of the maxilla and mandible. However, the activation area of skeletal Class â…¡ pattern was closer to the mandible, and that of skeletal Class pattern was closer to the maxilla…”

Is this a speculation based on a visual analysis or a quantitative analysis? Please, be specific in your comment, and if needed, use the word: we suggest that the main activation was …. (and cite Figure x)

  1. “…. The class 302 activation mapping technology indicates that the activation area of all CNNs were located 303 in the anterior jaws and lips. Most of the misclassified samples by CNNs were boundary 304 cases…”

I’m still missing information from the “mapping technology,” and you have also used this in your conclusion. Did you evaluate all the images one by one to be sure that the same region was used? And when I say if you evaluated, I’m asking if you have scientifically proven that it is the same region using some qualitative observation or quantitative. So please, if not, remove this statement.

Reviewer 2 Report

I found the study very interesting and certainly topical. AI systems will increasingly assist each clinician in his work, from the diagnosis to the therapy choice. 
